# Genomic Variations in Susceptibility to Intracranial Aneurysm in the Korean Population

**DOI:** 10.3390/jcm8020275

**Published:** 2019-02-25

**Authors:** Eun Pyo Hong, Bong Jun Kim, Steve S. Cho, Jin Seo Yang, Hyuk Jai Choi, Suk Hyung Kang, Jin Pyeong Jeon

**Affiliations:** 1Molecular Neurogenetics Unit, Center for Genomic Medicine, Massachusetts General Hospital, Boston, MA 02114, USA; ephong0305@gmail.com; 2Institute of New Frontier Stroke Research, Hallym University College of Medicine, Chuncheon 24252, Korea; luckykbj@naver.com; 3Perelman School of Medicine at the University of Pennsylvania, Philadelphia, PA 19104, USA; swcho92@gmail.com; 4Department of Neurosurgery, Hallym University College of Medicine, Chuncheon 24252, Korea; yiyaseo@naver.com (J.S.Y.); painsurgery@gmail.com (H.J.C.); nscharisma@daum.net (S.H.K.); 5Genetic and Research, Chuncheon 25252, Korea

**Keywords:** subarachnoid hemorrhage, delayed cerebral ischemia, genetics

## Abstract

Genome-wide association studies found genetic variations with modulatory effects for intracranial aneurysm (IA) formations in European and Japanese populations. We aimed to identify the susceptibility of single nucleotide polymorphisms (SNPs) to IA in a Korean population consisting of 250 patients, and 294 controls using the Asian-specific Axiom Precision Medicine Research Array. Twenty-nine SNPs reached a genome-wide significance threshold (5 × 10^−8^). The rs371331393 SNP, with a stop-gain function of *ARHGAP32* (11q24.3), showed the most significant association with the risk of IA (OR = 43.57, 95% CI: 21.84–86.95; *p* = 9.3 × 10^−27^). Eight out of 29 SNPs—*GBA* (rs75822236), *TCF24* (rs112859779), *OLFML2A* (rs79134766), *ARHGAP32* (rs371331393), *CD163L1* (rs138525217), *CUL4A* (rs74115822), *LOC102724084* (rs75861150), and *LRRC3* (rs116969723)—demonstrated sufficient statistical power greater than or equal to 0.8. Two previously reported SNPs, rs700651 (*BOLL*, 2q33.1) and rs6841581 (*EDNRA*, 4q31.22), were validated in our GWAS (Genome-wide association study). In a subsequent analysis, three SNPs showed a significant difference in expressions: the rs6741819 (*RNF144A*, 2p25.1) was down-regulated in the adrenal gland tissue (*p* = 1.5 × 10^−6^), the rs1052270 (*TMOD1*. 9q22.33) was up-regulated in the testis tissue (*p* = 8.6 × 10^−10^), and rs6841581 (*EDNRA*, 4q31.22) was up-regulated in both the esophagus (*p* = 5.2 × 10^−12^) and skin tissues (1.2 × 10^−6^). Our GWAS showed novel candidate genes with Korean-specific variations in IA formations. Large population based studies are thus warranted.

## 1. Introduction

Intracranial aneurysms (IAs), formed through the ballooning of intracranial arteries, are reported with an overall prevalence of 3.2% [1]. While annual rupture rates of up to 2% have been reported, subarachnoid hemorrhage (SAH) due to an aneurysm rupture is a life-threatening illness with a one-month mortality rate of up to 50% [2]. IAs represent a complex dysfunction with clinical implications and genetic components. Highly associated clinical risk factors for IA are sex (female), hypertension, and smoking [3]. The associations between IA and some heritable conditions, such as autosomal dominant polycystic kidney disease, Marfan syndrome, and Ehlers-Danlos syndrome, as well as family histories of IA suggest some possible genetic contributions to IA formation [4]. In particular, candidate loci such as 1p34.3-p36.13, 7q11, 19q13.3, and Xp22 have been reported in familial IA [4,5,6,7,8].

As of late 2008, previous genome-wide association studies (GWASs) have identified single nucleotide polymorphisms (SNPs) located within or near several candidate genes, including *EDNRA*, *SOX17*, *CDKN2A*, and *RBBP8*, which reached a genome-wide significance or suggestive threshold for the susceptibility to IA formation [9,10,11,12]. A meta-analysis of 61 studies in multi-ethnic populations further [13] demonstrated regions 4q31.23, 9p21.3, and 8q11 to be significantly associated with IA. Despite attempts to discover casual or candidate genes for IA formation or SAH development via GWAS-driven approaches, most of these studies have been conducted in both European and Asian-ancestry populations using the large-scale SNP array manufactured on the basis of Caucasian populations. Since the existence of ethnic difference should be considered in the fundamental identification of genomic susceptibility to complex diseases in multi-ethnic populations [14], even with in the same ethnicity, such as Japanese and Korean populations [15], the impact of susceptible loci on IA in the Korean populations remains to be clarified. 

Recently, The Axiom Asia Precision Medicine Research Array (PMRA), a comprehensive imputation-aware genotyping array on the basis of South- and East Asian populations, which contains more than 750,000 markers including disease-associated rare or common variations cataloged in public databases such as ClinVar and GWAS Catalog (Thermo Fisher Scientific, Waltham, MA, USA), was launched in South Korea and led to a new paradigm in genomic research. In this study, we aimed to identify IA-predicting genetic variations in Korean adults based on hospitalization records for SAH and IA. To the best of our knowledge, this study is the first GWAS report applying the Asian-specific PMRA chip to provide a better understanding of IA formation. 

## 2. Materials and Methods

### 2.1. Study Population

The study cohort was derived from the Chuncheon Sacred Heart Hospital stroke database (2015–2018). This database is a prospective and observational project in the regional medical center of the district of Chuncheon city, the capital city of the Gangwon Province in Korea [16]. Patients with a clinical diagnosis and neuroimaging consistent with hemorrhagic or ischemic strokes were recruited for the database. In this GWAS study, patients over 18 years of age with saccular and sporadic IA were analyzed. Non-saccular aneurysms, such as fusiform, dissection, traumatic, and infectious aneurysms, as well as familial aneurysms were excluded. Control subjects met the following criteria: 1) they underwent computed tomography or magnetic resonance angiography for headache evaluation or medical check-up; 2) had no concomitant neurological diseases, such as arteriovenous malformation, intracranial hemorrhage, or infarct; 3) had no familial history of IA or SAH in first-degree relatives; 4) had neither Parkinson’s disease (PD) or Alzheimer’s disease; and 5) were over 18 years of age [17,18]. Finally, this prospective study included radiologically confirmed IA with saccular shape (*n* = 250) and 296 controls enrolled from March 2015 to May 2018 at a single institution. Medical information was reviewed regarding sex, age, present symptoms (UIA or SAH), hypertension (HTN), diabetes mellitus (DM), hyperlipidemia, smoking, familial history of aneurysm, and the presence of SAH. Angiographic variables were reviewed concerning the aneurysm location (anterior vs. posterior) and size. This study was approved by the Institutional Review Board at the participating hospital (No. 2016-3, 2017-9). 

### 2.2. Genotyping and Sample Quality Control

Genomic DNA from peripheral blood was extracted using the HiGeneTM Genomic DNA Prep Kit (BIOFACT, Daejeon, Korea). The samples were genotyped using the Axiom^TM^ Asia Precision Medicine Research Array (PMRA) Kit (Thermo Fisher Scientific, Waltham, MA, USA), which contains over 750,000 SNPs including 50,000 novel markers covering East and South Asian populations based on the human genome version 19 (build 37). A total of 508,192 out of 798,148 variants, including small insertions/deletions (indels), passed the quality control test with a genotype call rate of ≥95%, minor allele frequency (MAF) ≥0.01, and Hardy-Weinberg equilibrium *p*-value ≥1 × 10^−6^, and these were tested in our GWAS.

### 2.3. Statistical Analysis

A descriptive analysis is shown using the numbers of subjects (percentage) for discrete and categorical variables, and means are shown with standard errors. We conducted a multivariate logistic analysis with seven confounding factors (sex, age, HTN, DM, hyperlipidemia, cigarette smoking, and four principal components) in 250 cases with IA and 294 controls after removing two missing values. These analyses were performed using STATA v11.2 (Stata Corp., Lakeway, TX, USA). Our genome-wide association analysis was conducted to identify candidate genetic variations for IA susceptibility under an additive inheritance effect model, after adjusting for ten potential covariates mentioned above using the PLINK 1.9 program (https://www.cog-genomics.org/plink/1.9/) [19]. We estimated the statistical power to evaluate a potential genetic reliability in GWAS using our candidate genetic variants demonstrating genome-wide associations with the Genetic Power Calculator (http://pngu.mgh.harvard.edu/~purcell/gpc/), based on the following assumptions: an IA prevalence of 5%, an odds-ratio (OR) in an additive effect model, a D-prime of 0.8, a 1:1.18 case-control ratio, and a type 1 error rate at a genome-wide significance level of 5 × 10^−8^. We further evaluated the genetic associations between 151 SAH and 99 UIA patients in a subsequent analysis. In addition, we analyzed single tissue expression quantitative trait loci (eQTL) to evaluate genetic effects on the expression levels of individual variants demonstrating genome-wide associations in our GWAS through the Genotype-Tissue Expression (GTEx v7) portal (https://gtexportal.org/home/) [20]. Manhattan and regional association plots were drawn by the “*qqman*” command in R package v3.4.0 (https://cran.r-project.org/web/packages/qqman) and LocusZoom v1.3 (http://locuszoom.org/), respectively [21].

### 2.4. Gene Functional Enrichment, Pathway, and Network Analyses

We analyzed the enrichment of gene ontology (GO) terms and Kyoto Encyclopedia of Genes and Genomes (KEGG) pathways, in order to functionally classify candidate genes for IA susceptibility with *p*-values less than 0.05 that were identified in our GWA analysis. For this we used the web-based program Database for Annotation, Visualization, and Integrated Discovery (DAVID) v6.8 (http://david.ncifcrf.gov/) [22]. Biological functions with a false discovery rate (FDR) of less than 0.01 were considered to be strongly enriched in the annotation categories. We displayed the functional association network of the identified proteins in the pathogenesis of IA formation using STRING v10 (http://string-db.org/) [23]. 

## 3. Results

### 3.1. Clinical Characteristics of the Enrolled Patients

The patients’ characteristics are summarized in Table 1. Among 546 patients, 250 patients (45.8%) were categorized as having IA. There were no statistical differences in gender distribution, HTN, DM, hyperlipidemia, and smoking between the two groups. Control subjects were younger than IA patients (*p* < 0.001). For IA patients, the incidence of SAH and UIA was 151 (60.4%) and 99 (39.6%), respectively. Anterior circulation aneurysms accounted for the majority of the IAs (*n* = 221, 88.4%), consisting of the middle cerebral artery (*n* = 71), the anterior communicating artery or the anterior cerebral artery (*n* = 59), the internal carotid artery (*n* = 38), or the posterior communicating artery (*n* = 53).

### 3.2. GWAS and Single-cell eQTL Analysis 

A total of 315 SNPs, including small indels, showed suggestive associations with *p*-values less than 1 × 10^−5^ in our GWAS after adjusting for confounder factors and four principal components in the enrolled patients (Figure 1 and Appendix A). 

After removing pairwise linkage disequilibriums of less than 0.8, 29 out of those SNPs achieved a genome-wide significance threshold *p*-value of less than 5 × 10^−8^ (Table 2).

The nonsense variant having a stop-gain function, rs371331393 located on *ARHGAP32* (11q24.3), showed the most significant association for IA formation (OR = 43.57, 95% confidence interval (CI) 21.84–86.95; *p* = 9.3 × 10^−27^) (Figure 2). 

In addition, four SNPs—rs75822236 (R535H, *GBA*, 1q22), rs112859779 (G141S, *TCF24*, 8q13.1), rs79134766 (A208T, *OLFML2A*, 9q33.3), rs3744644 (E639D, *SCARF1*, 17p13.3), and an additional nonsense SNP rs59626274 (stop-gained, *FLJ45964*, 2q37.3)—demonstrated potential direct relationships with coding proteins in sequences. The rs75822236 SNP showed the strongest associations with the risk of IA formation (OR = 161.46, 95% CI 53.86–483.60; *p* = 1.1 ×10^−19^). The “T” allele of this variant was rare in the control group compared with the “C” allele in patients with IA (MAF = 0.01 versus 0.33). The “T” allele in one splice acceptor and in transcript variant rs138525217 SNP located on the 3-prime untranslated region (UTR) of the *CD163L1* gene showed a rare occurrence and large effect size (MAF case/control = 0.31/0.01, OR = 75.98; *p* = 6.2 × 10^−23^). Eight out of 29 SNPs reached sufficient statistical power greater than or equal to 0.8, while other SNPs were underpowered in this GWAS because of the small sample size (Table 2). Two out of 17 previously reported SNPs—the rs700651 intron variant of the *BOLL* gene (2q33.1) and the rs6841581 located near the 5-prime UTR of *EDNRA* (4q31.22)—were validated in our study (*p* = 0.008 and 6.5 × 10^−4^, respectively) (Appendix A). Although none of the SNPs reached a genome-wide significance threshold for IA rupture, one intron variant, the rs6990241 of the *SGCZ* gene (8p22), showed a suggestive association with aneurysm rupture (OR = 0.34, 95% CI 0.22-0.55; *p* = 8.5 × 10^−6^) (Appendix A).

In a subsequent single-cell eQTL analysis on at least 70 samples from 45 human tissues, three SNPs showed significant differences in expressions (*p* < 1 × 10^−5^): rs6741819 (*RNF144A*, 2p25.1; *p* = 4.0 × 10^−14^) (Figure 3A) was down-regulated in the adrenal gland tissue (175 samples, effect size = −0.36, *p* = 1.5 × 10^−6^) (Figure 3B), rs1052270 (*TMOD1*. 9q22.33; *p* = 2.7 × 10^−8^) (Figure 3C) was up-regulated in the testis tissue (225 samples, effect size = 0.42, *p* = 8.6 × 10^−10^) (Figure 3D), and rs6841581 (*EDNRA*, 4q31.22; *p* = 6.5 × 10^−4^) (Appendix A) was up-regulated in both the esophagus (*mucosa*) (358 samples, effect size = 0.38, *p* = 5.2 × 10^−12^) (Appendix A) and the skin in areas not exposed to the sun (*suprapubic*) (335 samples, effect size = 0.21, *p* = 1.2 × 10^−6^) (Appendix A).

### 3.3. Gene Function and Network Analysis

In an analysis of gene function, 13 of 235 biological functions showed statistically significant GO terms (*p* < 0.05) (Table 3). 

Seven out of 31 IA-predicting genes—*FLJ45964*, *LINC01237*, *LINC00474*, *LINC02130*, *LOC102724084*, *NAPA-AS1*, and *SLC5A4-AS1*—were not available in either DAVID or STRING databases. Among those terms, the GO: 0044425 containing a “membrane part” function in the cellular component ontology was highly enriched in the cluster involving 15 genes (*p* = 0.0088, FDR = 9.7%). In particular, five genes, *EDNRA*, *LINGO2*, *GBA*, *DSCAM*, and *SCARF1*, played various roles in the development of cells, neurons, and cell membranes. A network analysis further supported the evidence of IA-predicting genes, including the three crucial genes *EDNRA*, *LINGO2*, and *GBA* (Appendix A).

## 4. Discussion

In this study, we investigated 29 novel genes and found 8 genetic variants demonstrating sufficient statistical power (above 80%)—*GBA*, *TCF24*, *OLFML2A*, *ARHGAP32*, *CD163L1*, *CUL4A*, *LOC102724084*, and *LRRC3*—and 5 variants with marginal statistical power (70% to 80%)—*RNF144A*, *FLJ45964, SPCS3*, *LINGO2*, and *MINK1*—for IA susceptibility. The *ARHGAP32* gene (also called p250GAP), containing the most significant nonsense mutation (rs371331393), regulates Rho GTPase activating protein and NMDA receptor signaling involved in RhoGAP. Rho GTPAases are associated with HTN through the regulation of vascular tone [24]. Bai et al. [25] showed Rho-specific GAP expression in smooth muscle cells in human and mice. ARHGAP42 knockout mice experienced significantly elevated blood pressure and cardiac contractility. In addition, the blood pressure increased significantly in response to endothelin-1 and angiotensin II [24,25]. Without a doubt, HTN is a risk factor for aneurysm formation and growth. Chronic HTN can cause intimal thickening with degeneration of the internal elastic lamina, causing a focal weakness of the arterial wall [26]. Thus, genes regulating Rho GTPase activity are likely involved in the pathogenesis of IA. Although the loci 11q.24.3 was statistically significant after adjusting for covariates including HTN, further study about the role of *ARHGAP32* gene in IA pathogenesis is required. 

Glucocerebrosidase (*GBA*) mutation is a well-known risk factor for PD. Malek et al. [27] reported that patients with pathogenic *GBA* mutations, including L444P, showed a 5 year earlier onset of PD than non-carriers. In addition, mild and severe heterozygous *GBA* mutations also affect the risk of PD and age of onset [28]. Lysosomal dysfunction, particularly lysosomal storage disorder due to lysosomal gene deficiency, is often a pathognomonic sign of neurodegenerative disease [29]. However, in ischemic or hemorrhagic strokes, the relationship to lysosomal dysfunction has not been well investigated, and prior studies have mainly focused on autophagic and lysosomal pathways after cerebral ischemia. Gu et al. [30] reported that autophagic bodies and lysosomes accumulated in neurons within 4 hours after an ischemic injury and gradually dissipated over time. Western blotting revealed increased levels of light chain 3-II and cathepsin B 4 hours after ischemic injury. Considering that matrix-degrading enzymes of lysosomal cysteine cathepsins are related to tissue remodeling and tumor invasion [31], further research on the impact of lysosomal mutation on IA pathogenesis is necessary. Also, the *MINK1* gene contributes to the mediation of mitogen-activated protein kinase signaling, as well as platelet activation and thrombus formation [32]. Intraluminal thrombi are known to contribute to aneurysm growth and rupture in abdominal aortic aneurysms (AAA), and thus, may play a role in IA as well [33]. Hansen et al. [33] measured platelet activation potential using computational fluid dynamics, but failed to demonstrate significant differences in stress-induced action potentials within aneurysm walls under resting or exercising conditions. Since platelet aggregation is more evident in smaller arteries [34] and IAs have a different mechanical flow pattern compared with abdominal aortic aneurysms, further study on the relationship between platelet activation and aneurysm formation is required. In addition, other novel gene variants should be investigated to identify their mechanism and clinical relevance to IA formation.

Abnormal extracellular matrix (ECM) remodeling and subsequent ECM degradation as well as endothelial injury in response to hemodynamic stress and growth factors, including transforming growth factor-β (*TGF β*), are related to IA formation and growth [35]. Connective tissue growth factor (*CTGF*, also known as *CCN2*) regulates vascular structural remodeling, cell proliferation, and differentiation. TGF β is a well-known regulator of *CTGF* expression. Ungvari et al. [35] suggested that ECM remodeling caused by high *CTGF* expression in the vascular wall is responsible for age-related large arterial pathogenesis such as atherogenesis and aneurysm formation. Although definite mutations in the *TGF β* receptors genes (*TGFBR1* and *TGFBR2*) were not noted in familial IA patients, some novel variants, including *TGFBR3*/Betaglycan, were noted [36]. Olfactomedin, like 2A (*OLFML2A*), one of the olfactomedin domain-containing proteins, play a role in cellular differentiation and is mediated by ECM components [37]. *OLFML2A* has a tissue-specific expression, without expression in neuronal tissue. The function of *OLFML2* protein has not been well characterized. Among ECM components, OLFML2 proteins bind to chondroitin sulphate-E and heparin preferentially [38,39]. Therefore, additional studies about the potential role of CTGF and *OLFML2A* in the process of ECM remodeling in IA development may be elucidating.

Endothelial injury is known to precede IA formation [40]. In such environments, clearance of apoptotic cells is important for tissue homeostasis, preventing morphological changes from a fusifom to a column shape [41,42]. Scavenger receptor class F, member 1 (*SCARF1*) binds to various endogenous (e.g., modified low-density lipoproteins, apoptotic cells, and heat shock proteins) and exogenous ligands of bacterial and viral antigens. *SCARF1* plays a crucial role in atherosclerosis, endothelial injury/dysfunction, inflammation, and host innate immunity, predominantly binding to carbamylated LDLs with its proatherogenic effect on human endothelial cells [42]. In our GWAS, mutations in *SCARF1* polymorphism, containing of common coding variant (rs3744644, E639D), showed a high correlation with IA formation, but this finding was underpowered. Therefore, the associations between *SCARF1* polymorphism and IA formation need to be studied in a larger population.

There exist genetic differences between inter-populations of the same ethnicity (i.e., Asians), and/or intra-population (i.e., Chinese populations) [43], as well as between different ethnicities, such as Asian and Caucasian populations [14]. In prior studies, Koreans showed different candidate SNPs for IA formation compared with the Japanese and Chinese population, as well as with the European population. Hong et al. [18] investigated four candidate SNPs located near or on the *SOX17* gene. Among them, only one SNP, the minor C allele of rs1072737 had significant association for IA in Koreans. With the elastin (*ELN*) gene, rs2071307 (Gly422Ser) was not associated with Korean IA, although a minor A allele of rs2071307 was significantly associated with Chinese IA [44]. We further evaluated previously reported SNPs in our results. Among the possible susceptible loci including 4q31.23, 9p21.3, and 8q11, which were mentioned in the previous reports [13], only two loci, rs700651 in the intron of *BOLL* [9] and rs6841581 near the 5-prime UTR of *EDNRA* [12], were replicated in our GWAS (Appendix A). Specifically, expression of the common susceptible “C” allele of the rs6841581 in the *EDNRA* gene, which is likely to inhibit transcription of EDNRA [12], was down-regulated in esophagus and skin tissues, and correlated with an increased risk for IA formation in our study. Interactions of endothelin-1 (EDN-1) and EDNRA were reported to function in IA pathogenesis and cerebrovascular vasospasms after SAH [45]. Elevated plasma levels of EDN-1 highly correlated with SAH patients developing sustainable vasospasms in the cerebrovascular inflammatory system [46]. In a subsequent GWAS, although none of our findings showed a genome-wide significance with SAH in patients with IA, we discovered that there was a suggestive association with the *SGCZ* gene, which is a component of the vascular smooth muscle sarcoglycan complex that contributes to muscular dystrophy pathogenesis [47] and should be further investigated for its contribution to SAH formation at the molecular level. 

In our study, transcription factor 24 (*TCF24*) was significantly related to IA formation, indicating protective effects. Associations between *TCF24* and IA formation have not been investigated before. *TCF24*, rs182838111 located at chromosome 8, increased the risk of Alzheimer’s disease via family meta-analyses [48]. In our analysis, patients diagnosed with Alzheimer’s disease were not enrolled as control subjects, although the potential inclusion of late-onset Alzheimer patients cannot be discounted. Accordingly, interpretation of GWAS, in particular involving *TCF24*, warrants caution. Different baseline characteristics, particularly patients’ age and DM, may affect aneurysm incidence. In our study, patients harboring IA were older with had a lower incidence of DM compared with control subjects. Age at diagnosis could be translated into the duration of follow-up period. A higher incidence of aneurysm, irrespective of aneurysm rupture, was noted in those over 75 [49]. Moreover, the patients’ age and aneurysm size were well correlated with F^14^C levels in SAH patients, indicating high turnover in collagen in the aneurysm walls [50]. Accordingly, additional studies regarding age stratification and IA formation would be necessary to further evaluate these relationships. Regarding DM in our study, it seemed to have a protective effect for IA formation (6.8% in IA vs. 12.5% in controls; *p* = 0.001). Dattani et al. [51] reported a negative association between AAA and DM. In addition, medications to treat DM can inhibit AAA formation and growth [51]. Therefore, further studies regarding DM and medications on IA formation and growth are needed.

There are some limitations to this study. Our GWAS enrolled 250 patients with IA and 294 controls from Korean adults selected from hospitalization and community-based cohorts. Our findings may have been underpowered to identify genetic variations for susceptibility to IA or SAH with the small sample size [52], even though some of the variants reached the genome-wide significance threshold with large effect sizes. Thus, further studies are warranted in a larger population-based cohort of patients and should be replicated in an independent cohort. In addition, our significant findings should be validated functionally at the molecular or cellular level to identify their roles in the multifactorial mechanical pathogenesis of IA formation. Finally, we did not thoroughly examine the patients who develop IA at a young age. Given the critical relevance of early age-of onset for IA, future studies in patients younger than 30 years will help elucidate important clues to the genetic architecture for IA occurring in the younger population. Despite these limitations, our findings may ultimately lead to the development of targeted therapeutic and preventive diagnosis for cerebrovascular diseases.

## 5. Conclusions

This GWAS is a powerful application of the Asian-specific PMRA chip that contains large-scale variations, including clinical and disease-associated markers. The evidence of novel candidate genes with Korean-specific variations suggest a genetic component to susceptibility to IA in Korean adults. Some of these variations, including *GBA*, *ARHGAP32*, *OLFML2A*, *SCARF1*, and *MINK1*, have been implicated to indirectly or directly contribute to IA pathogenesis through the inflammatory or cerebrovascular system in human blood vessels. Further study is necessary to validate these results in independent populations. Ultimately, a comprehensive approach integrating clinically relevant elements and genetic components may provide beneficial clues to evaluate the risk assessment and preventive medicine for the development of IA and subsequent rupture.

## Figures and Tables

**Figure 1 jcm-08-00275-f001:**
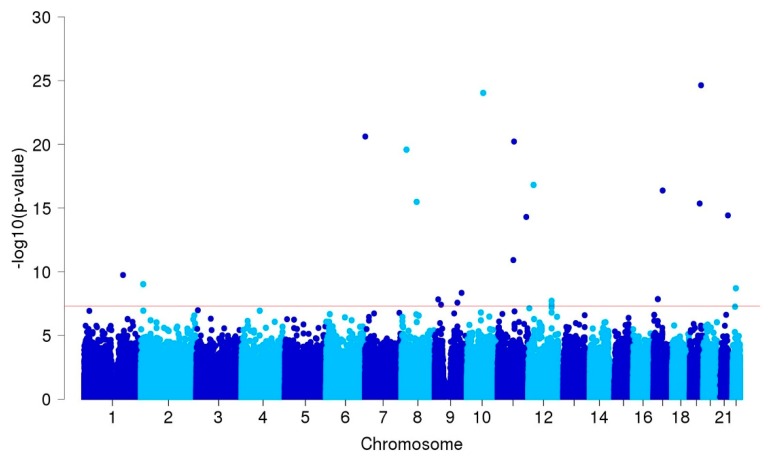
Manhattan plot of a genome-wide association study in 250 patients with intracranial aneurysms and 294 controls after adjusting for age, sex, hypertension, diabetes, hyperlipidemia, cigarette smoking, and four principal components. The red sub-line indicates the genome-wide significance threshold (*P* = 5 × 10^−8^).

**Figure 2 jcm-08-00275-f002:**
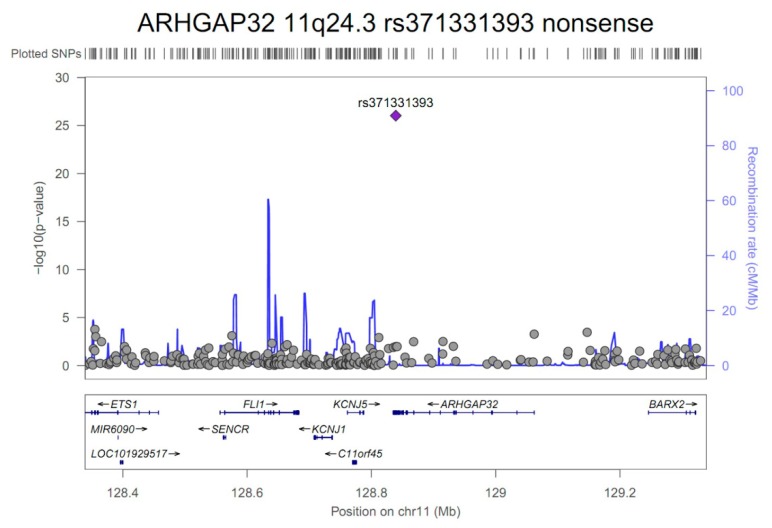
A regional plot of the rs371331393 (stop-gained mutation, *ARHGAP32*, 11q24.3) variant’s position ± 500 kb shows associations with intracranial aneurysms in 250 patients and 294 controls after adjusting for age, sex, hypertension, diabetes, hyperlipidemia, cigarette smoking, and four principal components. The x- and y-axes indicate the chromosomal position (mega base, Mb) and -log_10_ transformed *P*-values, respectively. The purple diamond indicates the rs371331393 variant demonstrating the most significant association in our genome-wide association study (*P* = 9.3 × 10^−27^). The gray circles indicate other SNPs within 500 kb of the rs371331393 variant’s position.

**Figure 3 jcm-08-00275-f003:**
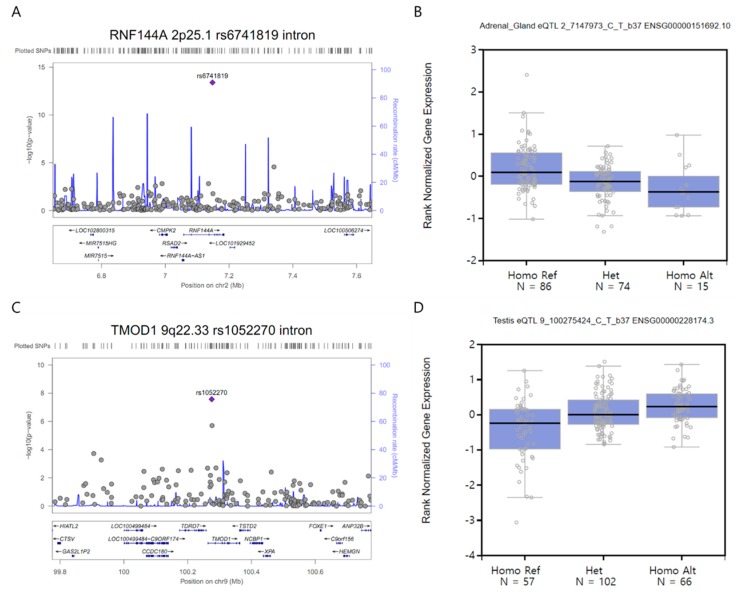
(**A** and **C**) A regional plot of (**A**) the rs6741819 (intron, *RNF144A*, 2p25.1) and (**C**) rs1052270 (intron, *TMOD1*, 9q22.33) variants (± 500 kb) demonstrates associations with intracranial aneurysms in 250 patients and 294 controls after adjusting for age, sex, hypertension, diabetes, hyperlipidemia, cigarette smoking, and four principal components. The x- and y-axes indicate the chromosomal position (mega base, Mb) and -log_10_ transformed *P*-values, respectively. The purple diamonds indicate the rs6741819 and rs1052270 variants with genome-wide associations in our study (*p* = 4.0 × 10^−14^ and 2.7 × 10^−8^, respectively). The gray circles indicate other SNPs within 500kb of the rs6990241 and rs1052270 variants’ position. (**B** and **D**) The expression level of (**B**) rs6741819 and (**D**) rs1052270 variants in the adrenal gland (*n* = 175) and testis (*n* = 225) tissues, respectively, according to single-cell expression quantitative trait loci (eQTL) in the GTeX portal (https://gtexportal.org/home). The x- and y-axes indicate three genotypes for each variant and the rank-normalized gene expression level, respectively. The Reference (Ref) and Alternative (Alt) alleles of each variant’s genotypes are equal to their major/minor alleles, respectively. Homo, homozygote; Het, heterozygote.

**Table 1 jcm-08-00275-t001:** Baseline Characteristics for Study Population.

Variables ^a^	Case (*n* = 250)	Control (*n* = 296)	*P* ^b^
Female, *N* (%)	146 (58.4)	154 (52.0)	0.74
Age, years	59.3 ± 0.8	52.1 ± 1.0	<0.001
Hypertension, *N* (%)	93 (37.2)	88 (29.7)	0.865
Diabetes mellitus, *N* (%)	17 (6.8)	37 (12.5)	0.001
Hyperlipidemia, *N* (%)	29 (11.6)	27 (9.1)	0.476
Cigarette smoking, *N* (%)	26 (10.4)	37 (12.5)	0.615
Principal components, mean ^c^			
Principal component 1	0.0014	−0.0012	0.169
Principal component 2	−0.0002	0.0001	0.656
Principal component 3	−0.0010	0.0008	0.039
Principal component 4	−0.0011	0.0009	0.031
Rupture status, *N* (%)			
UIA	99 (39.6)	-	-
SAH	151 (60.4)	-	-

SAH, subarachnoid hemorrhage; UIA, unruptured intracranial aneurysm. ^a^ Data are shown as the number of subjects (percentage) for discrete and categorical variables and mean ± standard error. ^b^
*P* values were estimated in the multivariate logistic regression model. ^c^ Principal components were estimated by performing the principal component analysis with four clusters.

**Table 2 jcm-08-00275-t002:** Genome-wide associations of 29 SNPs with intracranial aneurysm.

Gene	Chr.	SNP	Class	M/m ^a^	MAF ^b^	OR ^c^	L95 ^c^	U95 ^c^	*P* ^c^	Power ^d^
*PRDM2*	1p36.21	rs61775135	intron	C/A	0.16/0.41	0.32	0.23	0.43	3.6E-13	0.42
*GBA*	1q22	rs75822236	R535H	C/T	0.33/0.01	161.40	53.86	483.60	1.1E-19	1.00
*FMO4*	1q24.3	rs3737926	F281F	C/T	0.16/0.37	0.36	0.27	0.50	1.8E-10	0.32
*RNF144A*	2p25.1	rs6741819	intron	C/T	0.13/0.37	0.25	0.18	0.36	4.0E-14	0.72
*FLJ45964*	2q37.3	rs59626274	stop-gained	C/T	0.13/0.36	0.26	0.18	0.37	5.8E-14	0.71
*LINC01237*	2q37.3	rs78458145	intron	G/A	0.17/0.4	0.24	0.17	0.35	3.1E-15	0.68
*SPCS3*	4q34.2	rs17688188	intergenic	G/A	0.11/0.33	0.27	0.19	0.38	6.0E-13	0.73
*TCF24*	8q13.1	rs112859779	G141S	C/T	0.1/0.33	0.19	0.12	0.28	3.3E-16	0.94
*C9orf92*	9p22.3	rs12350582	intergenic	A/G	0.33/0.18	2.43	1.79	3.31	1.5E-08	0.20
*LINGO2*	9p21.1	rs56942085	intron	G/A	0.1/0.03	6.11	3.20	11.64	3.9E-08	0.78
*TMOD1*	9q22.33	rs1052270	intron	C/T	0.27/0.42	0.41	0.30	0.56	2.7E-08	0.15
*SUSD1*	9q31.3	rs79461840	intron	T/C	0.13/0.02	6.92	3.55	13.49	1.4E-08	0.65
*LINC00474*	9q33.1	rs4979583	intergenic	C/T	0.41/0.24	2.45	1.82	3.31	4.6E-09	0.24
*OLFML2A*	9q33.3	rs79134766	A208T	G/A	0.07/0.37	0.14	0.09	0.21	1.7E-19	0.96
*MYEOV*	11q13.3	rs76855873	intron	C/T	0.16/0.37	0.33	0.24	0.46	1.2E-11	0.43
*ARHGAP32*	11q24.3	rs371331393	stop-gained	G/A	0.32/0.02	43.57	21.84	86.95	9.3E-27	1.00
*CD163L1*	12p13.31	rs138525217	splice-site	C/T	0.31/0.01	75.98	32.13	179.70	6.2E-23	1.00
*SLC2A14*	12p13.31	rs118107419	intron	C/A	0.14/0.39	0.25	0.17	0.35	1.4E-14	0.67
*DRAM1*	12q23.2	rs7964241 ^e^	intergenic	A/G	0.35/0.21	2.49	1.81	3.42	2.0E-08	0.25
*CUL4A*	13q34	rs74115822	intergenic	G/A	0.19/0.03	6.25	3.58	10.90	1.1E-10	0.80
*LINC02130*	16p13.12	rs11646803	intron	C/T	0.28/0.48	0.45	0.35	0.58	4.8E-10	0.06
*LOC102724084*	16q23.2	rs75861150	intron	T/C	0.02/0.16	0.14	0.07	0.28	1.3E-08	1.00
*SCARF1*	17p13.3	rs3744644	E639D	G/C	0.18/0.35	0.39	0.28	0.54	6.0E-09	0.26
*MINK1*	17p13.2	rs72835045	intron	G/A	0.11/0.33	0.25	0.17	0.37	1.7E-12	0.79
*SLC47A1*	17p11.2	rs2440154 ^e^	intron	G/A	0.28/0.13	2.67	1.90	3.75	1.4E-08	0.26
*NAPA-AS1*	19q13.32	rs55800589	intron	G/C	0.32/0.4	0.38	0.29	0.49	3.4E-13	0.23
*DSCAM*	21q22.2	rs727333	intron	C/A	0.14/0.38	0.26	0.18	0.36	3.7E-14	0.69
*LRRC3*	21q22.3	rs116969723	P63P	G/A	0.11/0.35	0.24	0.16	0.34	3.8E-15	0.81
*SLC5A4-AS1*	22q12.3	rs117398778	intron	T/C	0.21/0.07	3.74	2.43	5.75	2.0E-09	0.56

Chr. chromosome; L95, lower 95% confidence interval; OR, odds ratio; U95, upper 95% confidence interval. ^a^ M/m, major/minor allele type; ^b^ MAF, minor allele frequency in case (left) and control (right); ^c^ OR, 95% confidence interval, and *p*-value were estimated in the multivariate logistic regression model after adjustment for age, sex, hypertension, diabetes, hyperlipidemia, cigarette smoking, and four principal components; ^d^ The statistical power for each SNP was estimated by performing the web-based Genetic Power Calculator (http://zzz.bwh.harvard.edu/gpc/cc2.html) using parameters, such as 5% prevalence of intracranial aneurysm, MAF in control, OR under an additive model, D-prime 0.8, 1:1.18 case-control ratio (296 controls/250 cases = 1.18), and type 1 error rate of 5e-8 (genome-wide significance threshold); ^e^ One SNP shown in a pairwise linkage disequilibrium (LD, *r*^2^ < 0.8): rs7964241 and rs56168082 (*r*^2^ = 0.81); rs2440154 and rs2289668 (*r*^2^ = 1.0).

**Table 3 jcm-08-00275-t003:** Gene enrichment functional analysis using candidate genes for susceptibility to intracranial aneurysms (*p* < 0.05).

Biological Function ^a^	*P* ^b^	FDR, % ^b^	Gene, *N*	Gene Set
*Biological process in ontology*				
GO: 0030154~cell differentiation	0.0117	16.8	10	*EDNRA*, *LINGO2*, *CUL4A*, *SLC2A14*, *MINK1*, *SCARF1*, *BOLL*, *DSCAM*, *TMOD1*, *GBA*
GO: 0031175~neuron projection development	0.0120	17.2	5	*LINGO2*, *MINK1*, *SCARF1*, *DSCAM*, *GBA*
GO: 0048468~cell development	0.0181	24.8	7	*EDNRA*, *LINGO2*, *MINK1*, *SCARF1*, *DSCAM*, *TMOD1*, *GBA*
GO: 0016358~dendrite development	0.0192	26.1	3	*MINK1*, *SCARF1*, *DSCAM*
GO: 0048869~cellular developmental process	0.0209	28.1	10	*EDNRA*, *LINGO2*, *CUL4A*, *SLC2A14*, *MINK1*, *SCARF1*, *BOLL*, *DSCAM*, *TMOD1*, *GBA*
GO: 0048666~neuron development	0.0210	28.3	5	*LINGO2*, *MINK1*, *SCARF1*, *DSCAM*, *GBA*
GO: 0007275~multicellular organism development	0.0260	33.8	11	*EDNRA*, *LINGO2*, *CUL4A*, *SLC2A14*, *MINK1*, *PRDM2*, *SCARF1*, *BOLL*, *DSCAM*, *TMOD1*, *GBA*
GO: 0051130~positive regulation of cellular component organization	0.0406	47.7	5	*LINGO2*, *CUL4A*, *SCARF1*, *DSCAM*, *GBA*
GO: 0030182~neuron differentiation	0.0454	51.6	5	*LINGO2*, *MINK1*, *SCARF1*, *DSCAM*, *GBA*
*Cellular component in ontology*				
GO:0044425~membrane part	0.0088	9.7	15	*FMO4*, *RNF144A*, *LINGO2*, *EDNRA*, *LRRC3*, *ARHGAP32*, *CD163L1*, *SLC2A14*, *SUSD1*, *SPCS3*, *MINK1*, *SLC47A1*, *DRAM1*, *SCARF1*, *DSCAM*
GO:0016021~integral component of membrane	0.0160	16.9	13	*FMO4*, *RNF144A*, *LINGO2*, *EDNRA*, *LRRC3*, *CD163L1*, *SLC2A14*, *SUSD1*, *SPCS3*, *SLC47A1*, *DRAM1*, *SCARF1*, *DSCAM*
GO:0031224~intrinsic component of membrane	0.0186	19.4	13	*FMO4*, *RNF144A*, *LINGO2*, *EDNRA*, *LRRC3*, *CD163L1*, *SLC2A14*, *SUSD1*, *SPCS3*, *SLC47A1*, *DRAM1*, *SCARF1*, *DSCAM*
GO:0016020~membrane	0.0205	21.1	17	*RNF144A*, *LRRC3*, *SUSD1*, *SLC2A14*, *MINK1*, *SLC47A1*, *SCARF1*, *FMO4*, *EDNRA*, *LINGO2*, *ARHGAP32*, *CD163L1*, *SPCS3*, *DRAM1*, *DSCAM*, *TMOD1*, *GBA*

FDR, false discovery rate; GO, gene ontology. ^a^ Functional categories in gene ontology; ^b^ Fisher’s exact *p*-value and FDR of each biological function were estimated by DAVID v.6.8 program.

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
