# Peer review of "Genomic Variations in Susceptibility to Intracranial Aneurysm in the Korean Population"

_jcm, 2019, doi:10.3390/jcm8020275_

Reviewer 1 Report

The authors investigated 29 novel genes and found eight genetic variants demonstrating sufficient statistical power and five variants with marginal statistical power for IA susceptibility. This paper is well-written, but there are several major issues that need to be addressed.

1.    Sample size may be too small to validate genomic variations statistically.

2.     Baseline characteristics between the two groups are different in several parameters, including age and with or without diabetes mellitus. These differences may affect the results.

Author Response

Comment 1: Sample size may be too small to validate genomic variations statistically.

Answer: Thank you for your comments on our manuscript. Sample size unfortunately is often an issue in genome-wide association studies (GWAS), which assume the 'Common Disease, Common Variant' hypothesis (References 1 and 2, below). To address this issue, we estimated the statistical power for genetic findings via the Genetic Power Calculator (http://pngu.mgh.harvard.edu/~purcell/gpc/). In our study, 13 out of 29 genetic variants showed sufficient statistical power. Nevertheless, our findings may be underpowered for intracranial aneurysm (IA) susceptibility genes due to the small sample size even though some of these variants had large enough effect sizes to reach the genome-wide significance threshold. Thus, further studies are warranted in a larger population-based cohort of patients and should be replicated in an independent cohort. For this study, we performed a GWAS replication study in an independent cohort to demonstrate susceptibility genes which we found initially in this study. We have added this information in this revised manuscript (page 11, line 336-340, highlighted). 

References

1.       Spencer C.C.; Su Z.; Donnelly P.; Marchini J. Designing genome-wide association studies: sample size, power, imputation, and the choice of genotyping chip. PLoS Genet. 2009,5:e1000477.

2.       Hong E.P.; Park J.W. Sample size and statistical power calculation in genetic association studies. Genomics Inform. 2012,10,117-122.   

Comment 2: Baseline characteristics between the two groups are different in several parameters, including age and with or without diabetes mellitus. These differences may affect the results.

Answer: As the reviewer pointed out, these six covariates, such as age, sex, history of diseases (i.e., diabetes mellitus (DM), hypertension, and hyperlipidemia), and cigarette smoking, are potential confounding factors that influence IA formation. Accordingly, in our GWAS, we made an effort to adjust for ten confounding factors, including these six covariates and four additional genetic ancestry variables of principal component factors to reduce potential bias due to the occurrence of population stratification on genetic variations in inter-populations. Nevertheless, different baseline characteristics, particularly patients’ age and DM may affect aneurysm incidence. In our study, patients harboring IA were older with lower incidence of DM compared to control subjects. Age at diagnosis could be translated into the duration of follow-up period. A higher incidence of aneurysm, irrespective of aneurysm rupture, was noted in those over 75 (Reference 1, below). Moreover, patients’ age and aneurysm size were well correlated with F14C levels in SAH patients, indicating high turnover in collagen in the aneurysm walls (Reference 2, below). Accordingly, additional studies regarding age stratification and IA formation would be necessary to further evaluate this relationships. Regarding DM in our study, it seemed to have a protective effect for IA formation (6.8% in IA vs. 12.5% in controls; p=0.001). Dattani et al. (Reference 3, below) reported a negative association between abdominal aortic aneurysm (AAA) and DM. In addition, medications to treat DM can inhibit AAA formation and growth (Reference 3, below). Therefore, further studies regarding DM and medications on IA formation and growth are needed. We have added this discussion in the revised manuscript (page 10, line 318-326; page 11, line 327-334, highlighted).

Overall, underlying differences in the patient demographics regarding age and DM could potentially affect our results, as the reviewer points out, but we tried to control for these differences to minimize potential biases.

References

1.      Asaithambi G.; Adil M.M.; Chaudhry S.A.; Qureshi A.I. Incidences of unruptured intracranial aneurysms and subarachnoid hemorrhage: results of a statewide study. J Vasc Interv Neurol. 2014,7,14-17.

2.      Etminan N.; Dreier R.; Buchholz B.A.; Bruckner P.; Steiger H.J.; Hanggi D.; Macdonald R.L. Exploring the age of intracranial aneurysms using carbon birth dating: preliminary results. Stroke. 2013,44,799-802.

3.      Dattani N.; Sayers R.D.; Bown M.J. Diabetes mellitus and abdominal aortic aneurysms: A review of the mechanisms underlying the negative relationship. Diab Vasc Dis Res. 2018,15,367-374.

Reviewer 2 Report

Genomic variations in susceptibility to intracranial  aneurysm in the Korean population

In general it is a nice study where novel SNPs are associated with intracranial aneurysm ruptures. However, a few issues have not been addressed.

In the introduction the following is stated:

“ Highly-associated clinical risk factors for IA are female sex, hypertension,  and smoking [3]. The associations between IA and some heritable conditions, such as autosomal  dominant polycystic kidney disease, Marfan syndrome and Ehlers-Danlos syndrome, as well as  family histories of IA suggest some possible genetic contributions to IA formation [4]. In particular,  candidate loci such as 1p34.3-p36.13, 7q11, 19q13.3 and Xp22 have been reported in familial IA [4-8].”

It has not been discussed in the discussion if any of these features can be confirmed in the Korean data. For instance, the smoking, hypertension and female sex do not. While, age and Diabetes do associate with IA. Moreover, Diabetes seems to protect against IA, which is also known for AAA. It is thought that metformin use is the cause of that, so that may be of interest as protective drug in the future for IA. There is an AAA trial starting soon with Metformin in the USA.

In addition, PKD is associated with mechanotransduction, since the cells lacking PKD 1or 2 do not sense flow (endothelial dysfunction probably), and Marfan Syndrome and Ehlers Danlos relation suggests ECM changes. So how do the found potential genes relate to mechanosensing/endothelial cells or ECM remodeling? For example, OLFML2A is an ECM protein, which may be of interest to focus on. It seems involved in CTGF signaling. And SCARF1 seems to be endothelial specific?

It is also of interest to mention in the discussion that this study does not find back the previously identified loci as mentioned below: “A meta-analysis of 61 studies in multi-ethnic populations further [13] demonstrated 4q31.23, 9p21.3, and 8q11 regions to be significantly associated with IA. ”

Although hardly anything is known about TCF24, in the human Gene Cards database, TCF24 is apparently also present in a GWAS study and related to age of onset of Alzheimers disease. I have not looked up what the cause of  Alzheimers disease was in this study. This may be relevant to mention.

What is the sentence in the abstract based on about inflammation. I do not see this reflected much in the data? Perhaps CD163L1, because of its expression on immune cells. Or perhaps GBA and SCARF1 because of lipid accumulation causing inflammation? This should be elaborated more. It did not stand out in the pathway analysis.

Author Response

Comment 1: In the introduction the following is stated: “Highly-associated clinical risk factors for IA are female sex, hypertension, and smoking. The associations between IA and some heritable conditions, such as autosomal dominant polycystic kidney disease, Marfan syndrome and Ehlers-Danlos syndrome, as well as family histories of IA suggest some possible genetic contributions to IA formation. In particular, candidate loci such as 1p34.3-p36.13, 7q11, 19q13.3 and Xp22 have been reported in familial IA.”

It has not been discussed in the discussion if any of these features can be confirmed in the Korean data. For instance, the smoking, hypertension and female sex do not. While, age and Diabetes do associate with IA. Moreover, Diabetes seems to protect against IA, which is also known for AAA. It is thought that metformin use is the cause of that, so that may be of interest as protective drug in the future for IA. There is an AAA trial starting soon with Metformin in the USA.

Answer: Thank you for your comments on our manuscript. In response to your recommendations, which highlight the same concerns as that of Reviewer 1, we have further discussed different characteristics of age and DM between aneurysm subjects and controls in the discussion section as follows:

Different baseline characteristics, particularly patients’ age and DM may affect aneurysm incidence. In our study, patients harboring IA were older with lower incidence of DM compared to control subjects. Age at diagnosis could be translated into the duration of follow-up period. A higher incidence of aneurysm, irrespective of aneurysm rupture, was noted in those over 75 (Reference 1, below). Moreover, patients’ age and aneurysm size were well correlated with F14C levels in SAH patients, indicating high turnover in collagen in the aneurysm walls (Reference 2, below). Accordingly, additional studies regarding age stratification and IA formation would be necessary to further evaluate this relationships. Regarding DM in our study, it seemed to have a protective effect for IA formation (6.8% in IA vs. 12.5% in controls; p=0.001). Dattani et al. (Reference 3, below) reported a negative association between abdominal aortic aneurysm (AAA) and DM. In addition, medications to treat DM can inhibit AAA formation and growth (Reference 3, below). Therefore, further studies regarding DM and medications on IA formation and growth are needed. We have added this discussion in the revised manuscript (page 10, line 318-326; page 11, line 327-334, highlighted).

Overall, underlying differences in the patient demographics regarding age and DM could potentially affect our results, as the reviewer points out, but we tried to control for these differences to minimize potential biases.

References

1.      Asaithambi G.; Adil M.M.; Chaudhry S.A.; Qureshi A.I. Incidences of unruptured intracranial aneurysms and subarachnoid hemorrhage: results of a statewide study. J Vasc Interv Neurol. 2014,7,14-17.

2.      Etminan N.; Dreier R.; Buchholz B.A.; Bruckner P.; Steiger H.J.; Hanggi D.; Macdonald R.L. Exploring the age of intracranial aneurysms using carbon birth dating: preliminary results. Stroke. 2013,44,799-802.

3.      Dattani N.; Sayers R.D.; Bown M.J. Diabetes mellitus and abdominal aortic aneurysms: A review of the mechanisms underlying the negative relationship. Diab Vasc Dis Res. 2018,15,367-374.

Comment 2: In addition, PKD is associated with mechanotransduction, since the cells lacking PKD 1or 2 do not sense flow (endothelial dysfunction probably), and Marfan Syndrome and Ehlers Danlos relation suggests ECM changes. So how do the found potential genes relate to mechanosensing/endothelial cells or ECM remodeling? For example, OLFML2A is an ECM protein, which may be of interest to focus on. It seems involved in CTGF signaling. And SCARF1 seems to be endothelial specific?

Answer: We appreciate your comments. In response, we have furtehre discussed OLFML2A and SCARF1 as follows:

Abnormal extracellular matrix (ECM) remodeling and subsequent ECM degradation as well as endothelial injury in response to hemodynamic stress and growth factors including transforming growth factor-β (TGF β) are related to IA formation and growth (Reference 1, below). Connective tissue growth factor (CTGF, also known as CCN2) regulates vascular structural remodeling, cell proliferation and differentiation. TGF β is a well-known regulator of CTGF expression. Ungvari et al. (Reference 1, below) suggested that ECM remodeling caused by high CTGF expression in the vascular wall is responsible for age-related large arterial pathogenesis such as atherogenesis and aneurysm formation. Although definite mutations in the TGF β receptors genes (TGFBR1 and TGFBR2) were not noted in familial IA patients, some novel variants, including TGFBR3/Betaglycan, were noted (Reference 2, below). Olfactomedin Like 2A (OLFML2A), one of the olfactomedin domain-containing proteins, play a role in cellular differentiation and is mediated by ECM components (Reference 3, below). OLFML2A has a tissue-specific expression, without expression in neuronal tissue. The function of OLFML2 protein has not been well characterized. Among ECM components, OLFML2 proteins bind to chondroitin sulphate-E and heparin preferentially (References 4 and 5, below). Therefore, additional studies about the potential role of CTGF and OLFML2A in the process of ECM remodeling in IA development may be elucidating (page 9, line 270-274; page 10, line 275-284, highlighted) .  

Endothelial injury is known to precede IA formation (Reference 6, below). In such environments, clearance of apoptotic cells is important for tissue homeostasis, preventing morphological changes from a fusifom to a column shape (References 6 and 7, below).  Scavenger receptor class F, member 1 (SCARF1) binds to various endogenous (e.g. modified low-density lipoproteins, apoptotic cells and heat shock proteins) and exogenous ligands of bacterial and viral antigens. SCARF1 plays crucial roles in atherosclerosis, endothelial injury/dysfunction, inflammation, and host innate immunity, predominantly binding to carbamylated LDLs with its proatherogenic effect on human endothelial cells (Reference 8, below). In our GWAS, mutations in SCARF1 polymorphism, containing of common coding variant (rs3744644, E639D), showed a high correlation with IA formation, but this finding was underpowered. Therefore, the associations between SCARF1 polymorphism and IA formation need to be studied in a larger population (page 10, line 285-295, highlighted).

References

1.        Ungvari Z.; Valcarcel-Ares M.N.; Tarantini S.; Yabluchanskiy A.; Fulop G.A.; Kiss T.; Csiszar A. Connective tissue growth factor (CTGF) in age-related vascular pathologies. Geroscience. 2017,39,491-498.

2.        Santiago-Sim T.; Mathew-Joseph S.; Pannu H.; Milewicz D.M.; Seidman C.E.; Seidman J.G.; Kim D.H. Sequencing of TGF-beta pathway genes in familial cases of intracranial aneurysm. Stroke. 2009,40,1604-1611.

3.        Anholt R.R. Olfactomedin proteins: central players in development and disease. Front Cell Dev Biol. 2014,2,6.

4.        Furutani Y.; Manabe R.; Tsutsui K.; Yamada T.; Sugimoto N.; Fukuda S.; Kawai J.; Sugiura N.; Kimata K.; Hayashizaki Y.; Sekiguchi K. Identification and characterization of photomedins: novel olfactomedin-domain-containing proteins with chondroitin sulphate-E-binding activity. Biochem J. 2005,389,675-684.

5.        Tomarev S.I.; Nakaya N. Olfactomedin domain-containing proteins: possible mechanisms of action and functions in normal development and pathology. Mol Neurobiol. 2009,40,122-138.

6.        Li M.H.; Li P.G.; Huang Q.L.; Ling J. Endothelial injury preceding intracranial aneurysm formation in rabbits. West Indian Med J. 2014,63,167-171.

7.        Ramirez-Ortiz Z.G.; Pendergraft W.F. 3rd.; Prasad A.; Byrne M.H.; Iram T.; Blanchette C.J.; Luster A.D.; Hacohen N.; El Khoury J.; Means T.K. The scavenger receptor SCARF1 mediates the clearance of apoptotic cells and prevents autoimmunity. Nat Immunol. 2013,14,917-926.

8.        Patten D.A. SCARF1: a multifaceted, yet largely understudied, scavenger receptor. Inflamm Res. 2018,67,627-632.

Comment 3: It is also of interest to mention in the discussion that this study does not find back the previously identified loci as mentioned below: “A meta-analysis of 61 studies in multi-ethnic populations further demonstrated 4q31.23, 9p21.3, and 8q11 regions to be significantly associated with IA.”

Answer: Previous GWAS and meta-analyses have been conducted in European ancestry populations (Reference 1 below). In our study, EDNRA (4q31.23) and BOLL (2q33.1), which were dicovered in these previous studies, were also found to be associated with IA. Per your recommendation, we have discussed this in the revision as follows:

There exists genetic differences diversity between inter-populations of the same ethnicity (i.e. Asians), and/or intra-population (i.e. Chinese populations) (Reference 2, below), as well as between different ethnicity, such as Asian and Caucasian populations (Reference 3, below). Among the possible susceptible loci including 4q31.23, 9p21.3, and 8q11, which were mentioned in the previous reports (Reference 1, below), only two loci, rs700651 in the intron of BOLL and rs6841581 near the 5-prime UTR of EDNRA, were replicated in our GWAS (Supplemental Table 2) (page 10, line 296-298; page 10, line 304-307, highlighted).

References

1.       Alg V.S.; Sofat R.; Houlden H.; Werring D.J. Genetic risk factors for intracranial aneurysms: a meta-analysis in more than 116,000 individuals. Neurology. 2013,80,2154-2165.

2.        Consortium HP-AS.; Abdulla M.A.; Ahmed I.; Assawamakin A.; Bhak J.; Brahmachari S.K.; Calacal G.C.; . Indian Genome Variation Consortium. Mapping human genetic diversity in Asia. Science. 2009,326,1541-1545.

3.        Huang T.; Shu Y.; Cai Y.D. Genetic differences among ethnic groups. BMC Genomics. 2015,16,1093.

Comment 4: Although hardly anything is known about TCF24, in the human Gene Cards database, TCF24 is apparently also present in a GWAS study and related to age of onset of Alzheimers disease. I have not looked up what the cause of Alzheimers disease was in this study. This may be relevant to mention.

Answer: In our study, transcription factor 24 (TCF24) was significantly related to IA formation, indicating protective effects. Associations between TCF24 and IA formation have not been investigated before. TCF24, rs182838111 located at chromosome 8, increased the risk of Alzheimer’s disease via family meta-analyses (Reference 1, below). In our analysis, patients diagnosed with Alzheimer’s disease were not enrolled as control subjects, although the potential inclusion of late-onset Alzheimer patients cannot be discounted. Accordingly, interpretation of GWAS, in particular involving TCF24, warrants caution. We have discussed this in the revision (page 10, line 318-323, highlighted).

Reference

1.        Herold C., Hooli B.V.; Mullin K.; Liu T.; Roehr J.T.; Mattheisen M.; Parrado A.R.; Bertram L.; Lange C.; Tanzi R.E. Family-based association analyses of imputed genotypes reveal genome-wide significant association of Alzheimer's disease with OSBPL6, PTPRG, and PDCL3. Mol Psychiatry. 2016,21,1608-1612

Comment 5: What is the sentence in the abstract based on about inflammation. I do not see this reflected much in the data? Perhaps CD163L1, because of its expression on immune cells. Or perhaps GBA and SCARF1 because of lipid accumulation causing inflammation? This should be elaborated more. It did not stand out in the pathway analysis.

Answer: Our findings, such as ARHGAP32, GBA, OLFML2A, SCARF1, and MINK1 genes, are likely to directly or indirectly affect inflammation and/or innate immune systems, which are crucial contributors to IA development (Reference 1, below). Therefore, we suggested that our main findings imply that IA pathogenesis lies in inflammatory interventions in the blood vessels of the cerebrovascular system. However, as you point out, our data did not directly  implicate inflammation as the main mechanism of IA formation in the Korean population. To better address this issue, we have been performing additional studies dealing with detailed functional mechanisms based on allele-specific strategy and coding variants in subsequent in silico functional study. To avoid misunderstanding, we have revised the conclusion of our abstract as follows: 

Our GWAS showed novel candidate genes with Korean-specific variations in IA formation. Large population-based studies are warranted.

Reference

1.      Chalouhi N.; Ali M.S.; Jabbour P.M.; Tjoumakaris S.I.; Gonzalez L.F.; Rosenwasser R.H.; Koch W.J. Dumont AS.Biology of intracranial aneurysms: role of inflammation. J Cereb Blood Flow. 2012,32,1659-1676.